# Renewable Electricity Transition: A Case for Evaluating Infrastructure Investments through Real Options Analysis in Brazil

Anna Carolina Martins [1,*], Marcelo de Carvalho Pereira [1] and Roberto Pasqualino [2]

1 Institute of Economics, University of Campinas, Campinas 13083-970, Brazil; mcper@unicamp.br
2 Department of Land Economy, University of Cambridge, 19 Silver Street, Cambridge CB3 9EL, UK; rp747@cam.ac.uk
* Correspondence: annacarolmartz@gmail.com

**Abstract:** This paper explores the uncertainty of expected returns by adopting the real options analysis method for the financial evaluation of renewable energy projects in Brazil. Energy transition is key to meeting climate targets, and real options analysis can play a pivotal role in evaluating renewable energy projects to meet those targets. The impact of the volatility of the chosen variables on the viability of the project is studied using Monte Carlo simulation in the R software. The results indicate that the lower the option value the higher the volatility of the project, leading to lower likelihood of the project being financed. The resulting model represents a simple instrument that can be incorporated in larger modelling frameworks (e.g., agent-based simulation) to assess the impact of real option analysis on different markets and environmental and socio-political conditions. These findings represent a strong case for the adoption of systems modelling to inform policy to support global energy transition, as the application of this method can make a renewable energy project financially more attractive in comparison to those relying on carbon intensive energy sources.

**Keywords:** renewable energy; finance; volatility; real option analysis; Monte Carlo Simulation

## 1. Introduction

The most widespread electricity generation technologies are based on non-renewable energy sources, leading to high volumes of anthropogenic greenhouse gases (GHGs) dispersed in ecosystems [1] One of the reasons for this result is the competitive cost of generating electricity using non-renewable energy that still represents a barrier for the widespread diffusion of renewable energy in developing countries [2].

According to data from the International Energy Agency (IEA) [2], the main source of energy worldwide is oil, followed by coal, gas, hydro, nuclear, wind, biofuel, solar (photovoltaic and thermoelectric), waste, geothermal, and ocean. However, electricity production from non-renewable sources has been declining in recent decades [2]. Some of the reasons for the recent improvement have been the new environmental policies and the increased efficiency of renewable electricity (REE) options. However, these reductions are modest, requiring increasing efforts in establishing the use of sustainable alternatives to build capacity in the medium and long term to achieve climate targets in this decade [2,3].

Despite the consensus that a quick energy transition is needed, the current levels of private funding for "green" energy are still insufficient [4–6], and it is unlikely that public investments alone will reach the necessary levels, given the huge investments required [7].

In Brazil, for instance, the situation is not an exception. Public banks such as the National Bank for Economic and Sustainable Development (BNDES) and the Bank of Brazil (BB) are the largest funders of renewable energy [8,9], and another substantial part of investments is made by international development banks such as the European Investment Bank (EIB) and the Agence Française de Développement (AFD). Investments made by the

private sector are insufficient and reliant on public policies and incentives [10,11]. In fact, one of the current challenges is to stimulate the implementation of REE plants with a good synergy between the private and public sectors. A critical issue is the adequate allocation of financial resources for REE investments ensuring that the financing of transition projects from non-renewable electrical energy (NREE) to REE is feasible.

One major bottleneck in private green energy finance is the scarcity of REE investments [12]. The most common project evaluation practice is based on discounted cash flows based on current market expectations (also known as net present value (NPV)) and considers funds required for both capital and operational expenditures (CAPEX/OPEX). However, this project evaluation method does not consider that the parties involved could have the right of changing their strategies as the project develops. This means that they must keep the committed resources dedicated to the project after the financing contract is stipulated, even if the evolution of the expectations changes the project valuation significantly in due course. Given that green energy projects may require large financial resources over long periods, such contractual rigidity may easily discourage private sector investment, in particular, due to the significant uncertainty about some key expected values required by the NPV calculation. For instance, it can be trivially demonstrated that energy prices exhibit significant volatility in the long run, and, in turn, do the expected project revenues and valuation. Considering the uncertainty involved in the many variables employed in a NPV valuation, this leads to creditors requiring higher (internal) rates of return for the projects to be financed, potentially discarding many projects that would prove to be perfectly viable with ex post assessment.

This paper proposes a system modelling application with the purpose of supporting building evidence that promotes the green transition agenda in Brazil. The model core component is in the adoption of a financial evaluation method for renewable energy projects that incorporates the uncertainty of expected returns for the case of the Brazil economy. To model the decisions of the agents involved, we will assume that agents, financiers, and entrepreneurs behave strategically, relying on real options analysis (ROA). ROA is a complement to the investment analysis performed using the Net Present Value (NPV) criteria by including the uncertainty of cash flows and exploring opportunities to change investment decisions in the analysis [13]. One of the main contributions of this paper is to explore the importance of considering the value of such strategical opportunities, if any, when evaluating the NPV, in particular when the uncertainty associated with the expected values of key variables is high, as in the case of REE projects. Additionally, we incorporate the uncertainty associated with variables such as wages and labour productivity in the ROA analysis as a complement to what has been discussed over the years by authors such as Kjaerland [14], Batista [15], Zavodov [16], and Kim [13].

Specifically, Kjaerland [14] showed that the application of real options was adequate to explain the aggregate investment in hydro power in Norway. Batista [15], in turn, showed that the expected value based on real options became superior to the traditional NPV approach by incorporating the possible strategical flexibilities in the development of a project. Zavodov [16] engaged in discussing how the application of real options, especially in hydro power projects, could be efficient in developing economies. Kim [13] proposed a framework for evaluating renewable energy investment based on real options. However, none of them considered the importance of variables such as wages and labour productivity. Volatility on the forecast of essential factors, such as wages, productivity, (imported) equipment prices, and exchange rates, has been historically high, particularly in developing countries.

The Section 2 of the paper presents the literature review from the perspective of decision making, project evaluation, and finance. The Section 3 describes the proposed valuation model, and the Section 4 applies the model to a hydro power plant in Brazil, including sensitivity analysis based on real data. Section 5 discusses the findings, and Section 6 concludes the paper.

## 2. Literature Review

### 2.1. Wider Landscape for Global Energy Transition

There is increasing evidence that indicates how changes in the physical and biological systems relate to the increase in the accumulation of greenhouse gas (GHG) emissions into the atmosphere. The increase in global average temperature has proven to be a significant factor helpful to measure such changes and shown by more frequent extreme weather events in different regions of the world [17]. According to Loureiro (2019), the development of a global economy that favours the energy transition was marked by four events: (1) the enactment of the American environmental policy in 1969 (NEPA), (2) the United Nations conference in Stockholm in 1972, (3) the publication of the report "Our common future" in 1987, and (4) the United Nations conference in Rio de Janeiro in 1992 (Rio-92) [18].

To monitor the progress of the commitments made in Kyoto [19], a series of conferences were held up until 2012. In 2015, at COP-21 (21st Conference of Parties), the "Paris Agreement" was approved, and it was determined that the increase in the planet's average temperature should not exceed +2 °C above pre-industrial levels. However, it was only at COP-22 that rules were defined to enable the fulfilment of the Paris Agreement. The most recent conference COP-27, in 2022 in Egypt, among other key points, suggested that the new climate target should be limited to +1.5 °C of global temperature increase in comparison to preindustrial levels. Although 1.5 °C may seem a non-significant increase in the global north, it may be categorical for the future existence of some coastal cities in the global south [20].

As IEA [21] highlights, the power generation sector sustains modern society by supplying energy but is also responsible for most of the GHG emissions. Thus, there is enormous potential for climate change mitigation related to the reduction in GHG emissions associated with power generation. In 2020, power generation was responsible for 40% of $CO_2$ emissions worldwide, making the focus on alternative technologies evident [22].

According to the United Nations Renewable Energy Observatory for Latin America and the Caribbean (UNIDO), among the options for renewable electricity energy (REE) generation we have (1) hydropower, (2) geothermal, (3) wind, (4) ocean, (5) solar, and (6) biomass [23]. As the IEA [2,24,25] report shows, the diversity of renewable sources in power generation has transformed the sector globally and the advancement of REE generation sources has outpaced the growth in electricity demand [21]. Despite this, the investment in these projects remains largely affected by uncertainty due to the impact of the decisions of the agents involved, which can be linked to both macroeconomic and microeconomic variables [26,27].

### 2.2. Accounting for Uncertainty in Project Evaluation

From a macroeconomic perspective, uncertainty is primarily linked to the possibility of changes in regulatory policies that might alter the conditions of the REE generation market, affecting, for instance, variables such as interest, exchange rates, or employment levels [28]. From a microeconomic viewpoint, uncertainty may be related to factors such as the scarcity of raw materials, technical difficulty, availability of skilled labour, and volatility of demand for electricity, among others [29].

If the uncertainty is related to the possibility of regulatory changes that may affect the interest and exchange rates, the agents involved can use the contract as a mitigating tool for this uncertainty. The level of employment and disposable income in an economy directly impact energy consumption but cannot be stipulated in a contract [30,31]. In this case, agents' decisions and strategies end up being guided by long-term expectations about the behaviours of these variables. These expectations, as Carvalho [32] pointed out, must be supported by estimates based on past and current performance.

When related to the availability of raw materials, uncertainty in REE generation projects is also associated with climate change. As Barbosa [29] shows, the raw materials of REE come from natural cycles that are currently abundant, although distributed in different proportions on the globe. Despite this, these sources of electricity are subject to

the conditions of nature. Thus, a change in climate patterns could affect the production of REE for a long period, with direct implications for the price of REE.

The price of electrical energy (EE) (usually charged per unit of energy In kilowatt per Hour-kWh) is inclusive of the costs incurred in the process of generation and distribution to consumers and includes charges and taxes. In addition, given the essentiality of EE, its full-time availability is also included in the price. In this sense, both the scarcity of raw material and any technical difficulty that generate uncertainty about the expected result of the project affect the price of energy [33].

In general, we can summarize the uncertainties in the project development in two groups: (1) the uncertainty arising from the risks directly associated with the project performance and (2) the uncertainty arising from the risks linked with the country's business environment [34] In the first group, De Araújo [35] distinguished the construction, operation, and financial risks, i.e., construction delay or abandonment or unexpected cost increase, choice of inappropriate technology, environmental risk, wrong estimates, lack of inputs, consumer market, inadequate product price, extremely high interest rates, and exchange rate risk. In the second group, specific economic, political, social, and geographical characteristics are associated. For example, regulatory institutional changes, tariff adjustments, tax changes, sudden changes in monetary, fiscal or exchange rate policies, uncontrolled public deficit, or private debt [35].

As a result, the development of a project financing is not able to abstain from a certain level of uncertainty. This is because even when using approaches such as Project Finance, a financing modality directed to the implementation of large infrastructure projects, the existence of distinct stages in the development of the project ends up spreading the uncertainty on several factors or agents. In this paper (see Section 3), we are considering uncertainty from the perspective of both groups, highlighting the possibility of oscillations in variables such as energy prices or tariff, wages, utilisation, and labour productivity.

### 2.3. The Implementation Stages of Renewable Energy Projects

Schematically, the implementation of an REE project can be summarised in three stages, as shown in Figure 1: the first stage consists of the design and evaluation of the project, i.e., the elaboration of the Base Project; the second stage contemplates the execution of the Base Project, e.g., the construction of the power plant; the third stage refers to the operationalization of the plant.

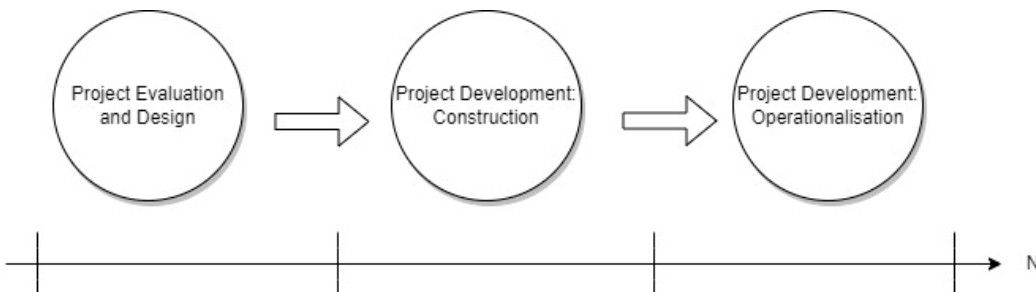

**Figure 1.** Summary of EER project implementation steps. Source: Own elaboration.

In the first stage, there must be a study about the location and the community that will be served by the project [36]. This is because the patterns of electricity demand differ based on the geographical location and cultural habits. In addition, the economic structure of the community should be evaluated due to the profile of existing economic activities (such as farming, industry, tourism, and services) can affect the way of producing energy efficiently [36]. In regard to the technical aspects, the first stage includes the characterization of the power generation system, which can be split between autonomous and grid connected. Autonomous systems should be able to respond to demand peaks independently indifferently from grid-connected systems [37]. In summary, the first stage in

the preparation of an REE project consists of a detailed definition of the electricity demand, the available generation possibilities, and the assessment of the negative impacts caused by the plant in targeted region [38].

The second stage of the project, the construction of the REE plant, represents the largest expenditure for project implementation. For this reason, in many hydroelectric projects, the viability of the plant construction is evaluated through the Gibrat ratio [39]. In this case, the smaller the ratio between the length of the dam and the energy production capacity the greater the feasibility. At this stage, the environmental impacts arising from the construction of the plant, whether related to fauna or flora, must be mitigated.

The last stage (operationalization) is usually supervised by the responsible organisation in the country. In Brazil, the federal agency responsible for the coordination and control of the operation of power generation plants is the National Interconnected System (SIN), and the planning of the operation of isolated systems, under the supervision and regulation of the National Agency of Electric Energy (ANEEL), is the National Electric System Operator (ONS) [40].

Although important, SIN, ANEEL, or ONS do not decide which projects will be financed or not. As mentioned, their scope of action is to coordinate, supervise, and regulate, respectively. As explained in the following section, the evaluation is an economic process.

*2.4. Economic Evaluation of REE Generation Projects via ROA*

Historically, infrastructural investment in Brazil has relied mainly on public sources of capital [41]. Even if precise figures are not available, there is evidence that the role of private creditors in renewable energy finance is mainly as a mediator of public institutions' programs, such as National Bank for Economic and Sustainable Development (BNDES) Fundo Clima or Finem "indirect support" modality or BNDES Garantia and FGI credit guarantee instruments [9,42].

The risk aversion of public entities (like BNDES) in requiring substantial credit guarantees from candidate projects may be justified by the Brazilian regulatory framework [42]. However, such constraints may not apply to private creditors when supplying their own funds. The understanding of the regulatory conditions that can influence the behaviour of (real) private finance of renewable and increase their support to the green transition appears to be a key issue to be addressed by policy analysts and decision makers. Supporting a regulatory framework that enables new financing agents and models is crucial for a successful green transition, in particular in developing countries where credit for investment has been historically scarce and expensive, such as Brazil [9].

The process valuation of a project includes the identification and quantification of the benefits and harms attributable to its implementation over a given period. A number of methodologies has been proposed in the literature seeking to assign or establish feasibility parameters [43]. Among the commonly used methods, there are the discounted cash flow, also known as net present value (NPV), and, more recently, the real option analysis. Real option analysis (ROA) has been increasingly used in the evaluation of renewable energy projects [44–46], and it can be seen as a complementary part of the traditional evaluation process.

Unlike usual financial options, where the underlying assets are liquid assets (easily traded), real options are applied to real assets such as investment projects. The key idea is that the parties involved (i.e., creditor and developer) may change their decisions about the financing and development of a project after it started, without incurring in a breach of contract or litigation [47]. There are distinct types of Real Options, Gazheli [48], highlights:

○ Postpone: possibility to wait to invest in the project. Thus, the irreversible investment may happen only when more information of future market and production conditions is available.
○ Abandon: opportunity to abandon the project and take any residual value back.
○ Alter: flexibility to change the project, through the possibility of altering the form of production, given future market and production conditions.

When receiving finance requests, and where the economic environment or the future context is uncertain, the creditor may wish to wait a certain period before deciding on whether to invest in a renewable energy project. In such a case, the deferral option offers the chance to participate in such projects at some point in the future. The change option, on the other hand, would enable the agents to switch to technologies or business models that prove to be more profitable over time. Or, if a project offers different ramifications such as wind, solar, or hydro, the investing agent can acquire the right to switch between technologies according to the market feasibility.

The exit option provides both agents (creditor and developer) the right, but not the obligation, to leave the project before the fixed term, i.e., if for any reason the project's financial performance is affected negatively, both the creditor and the developer may (within an agreed period) opt to exit the credit operation, therefore, allowing opportunities to change investment decisions. As a result, the creditor may abstain from the obligation to finance other stages previously established in the contract, both agents must agree on the period(s) in which they can exit. Furthermore, the exit option does not exempt the developer from paying the amount already borrowed off. Therefore, the exit option provides an alternative instrument to reducing the risk taken by the creditor.

The valuation of real options in investment projects can be computed using different methods, as shown in Table 1. According to Marques [49], the binomial lattice method emulates the option valuation method presented by Black and Scholes [50], and, as it does not require tractable statistical models, it allows for far greater flexibility on the option formats that can be valued.

**Table 1.** Real options analysis methodologies applied to renewable energy projects (in chronological order of completion).

| Authors | Country | Type ER | Methodology |
|---|---|---|---|
| HOFF [51] | California | Photovoltaic | BT |
| ZHANG X [52] | Non-Regional | Hydraulics | SIM |
| KJAERLAND [14] | Norway | Hydraulics | PDE |
| KUMBAROGLU [53] | Turkey | Wind | PDE |
| LEE [46] | Taiwan | Wind | BT |
| YANG [54] | China | Wind | SIM |
| BATISTA [15] | Brazil | Hydraulics | SIM |
| LEE [45] | Taiwan | Wind | EA |
| ZAVODOV [16] | China | Hydraulics | EA |
| REUTER [55] | England | Wind | PDE |
| BOOMSMA [56] | R. Nordic | Wind | AMMQ |
| LEE [44] | Indonesia | Hydraulics | GT + SIM |
| KRONIGER [57] | England | Wind | PDE + SIM |
| KIM [58] | Korea | Wind | BT |
| ABADIE [59] | UK | Wind | PDEPDE |
| WEIBEL [60] | England | Wind (onshore and offshore) | AMMQ |
| JEON [61] | Korea | Photovoltaic | MP |
| ZHANG [62] | China | Photovoltaic | PDE |
| KIM [13,63] | Korea | Hydraulics | PDE |
| AGATON [64] | Philippines | ER | PDE + AMMQ |
| GAZHELI, [48] | Non-Regional | Solar and Wind | PDE |

Legend: PDE = Partial Differential Equations; BT = Binomial Tree; SIM = Simulation; PD = Dynamic Programming; EA = Empirical Analysis; AMMQ = Monte Carlo Least Squares Approach; GT = Game Theory; MP = Probabilistic Model. Source: Own Elaboration.

The application of the binomial tree method consists of two stages. The first consists of the project valuation and, consequently, the application of Equations (1) and (2). The second stage consists of valuing the project option and applying Equations (3) and (4).

To construct the binomial valuation tree for a project whose investment is $S_0$ at the initial point of valuation ($n = 0$), we start a binary tree (two branches starting from each node) with the root node at $n = 0$. At each $n$ subsequent period, the value $S_n$ of each node, relative to the value of the option to defer the investment decision from 0 to $n$, is unfolded into two new nodes, relative to the time point $n + 1$. $N$ represents the total number of decision moments considered in the calculation. Decision moments do not necessarily represent linear units of time ($t$) but only a sequence $n = 0,1, \dots , N$ relative to each moment in which a decision on the investment may be made or valued by the agents.

The value of the project in each of the two new nodes of the tree is obtained by multiplying the value of the previous node $S_{n-1}$ by the risk factors $\phi_u$ e $\phi_d$. Figure 2 presents the binomial tree for a three-step decision process ($N = 3$). The value *SI* at node I, for example, can be obtained by the product $S_0\phi_u\phi_d^2$ the value at node I, which, for example, can be obtained by the product obtained by the ABEI path or by the numerically equivalent ACEI or ACFI paths. The paths represent the various possibilities of project development, given the uncertainty. The products represent how the additional information is incorporated in the project valuation. Thus, each node represents the future values of the project and is positioned at a constant logarithmic distance. This means that the stochastic value of the project in continuous time and state can be approximated by a discrete process and state.

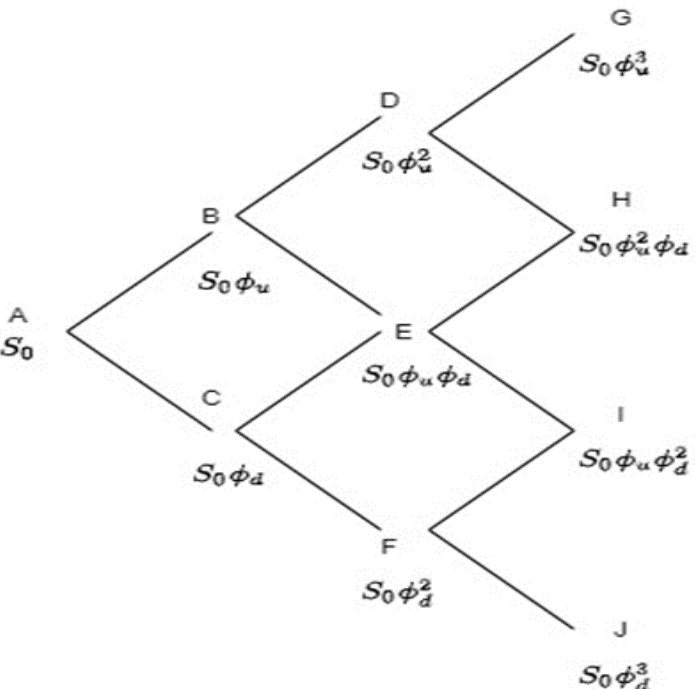

**Figure 2.** Example of binomial tree of project valuation with three decision steps.

After constructing the binomial tree of project valuation, the options valuation tree of the model can be developed. To this end, the backward induction method proposed by Kim [58] was applied. This method proposes that the values of the last and intermediate nodes of the option valuation tree are obtained by subtracting the difference between the values in the project valuation tree and the initial investment.

To calculate the value of each option $V_n$ recursively, we discount from the value of subsequent nodes $V_{n+1}^{u,d}$ the risk-free rate $r$ weighted by the risk-neutral probability $q$. Figure 3 presents the binomial option valuation lattice for a three-step decision process ($N = 3$).

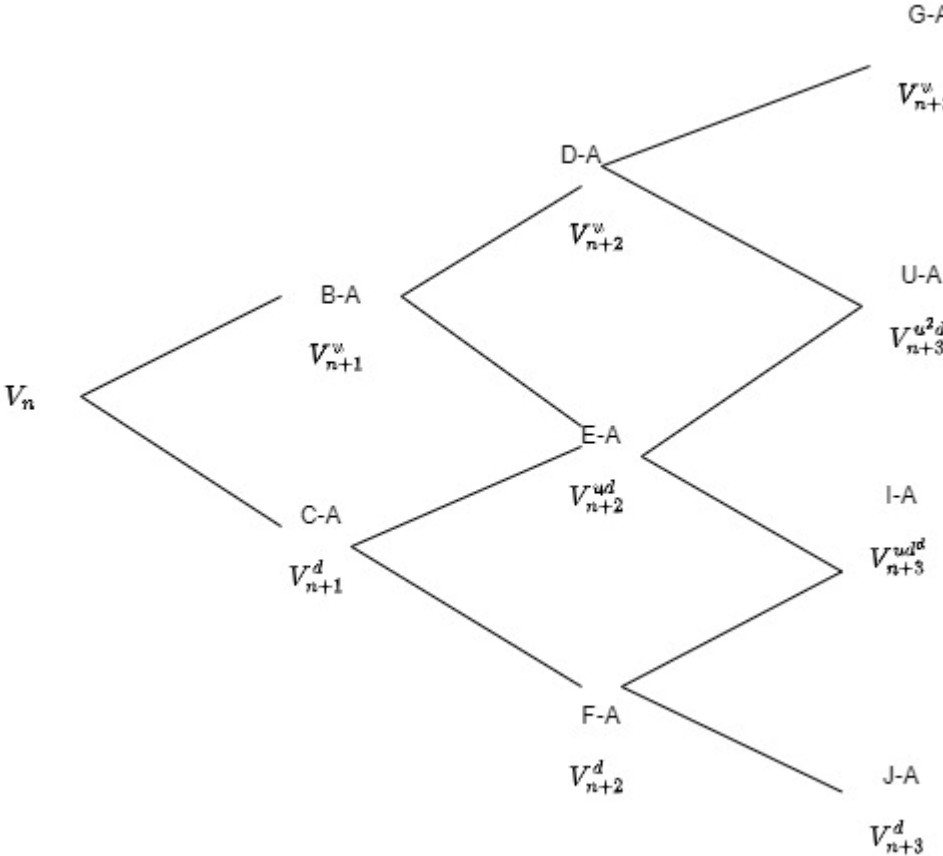

**Figure 3.** Option Valuation Tree.

The decision tree model provides the investor with an overview of the investment, highlighting alternatives and option values and reducing project risks. According to De Castro Rodrigues and Rozenfeld [65], this method is used mainly when asset values cannot be determined analytically. When there are no "closed" mathematical formulas that describe the researched phenomenon, other models are required, such as the binomial lattice and Monte Carlo simulation.

*2.5. Discounted Cash Flow and Strategic Net Present Value*

The discounted cash flow (DCF) of the project is obtained from the sum of the difference between income and expenses in each period, duly adjusted by the time value of money (interest rate). The DCF consists of the projection of the project's future results, adjusted for a single period. The discount rate applied is the weighted average cost of capital (WACC) which, according to Fernandez [66], reflects the project's capital cost structure.

Strategic net present value (SNPV) is an adaptation of the discounted cash flow method and the concept of net present value (NPV) to the context of decisions that can be postponed. The SNPV includes the option value of being able to abandon or modify the project after its start in the analysis [13].

According to Silva [67], in calculating NPV, flows are intrinsically fixed since all decisions are taken simultaneously in $n = t = 0$ and are defined as the difference between the present values of revenues and expenses throughout the project. As Abdelhady [68] explains, NPV is an indicator of the economic viability of a project. A positive NPV indicates that the project is viable at the time of analysis, considering the information available thus far.

When considering the option value (of deciding in the future), as presented above, agents incorporate the value of potential future decisions into the analysis, which could mitigate losses in the unfortunate scenario where the project does not develop as expected.

According to Kim [13], the use of SNPV as a project evaluation method is a superior alternative to NPV for projects with uncertainty in their main variables, as is the case of REE. For example, REE projects have a volatile cash flow due to the uncertainty in future market conditions that imply considerable risk, for which the possibility of abandonment has a real value that needs to be considered in the project evaluation.

The logic for comparison of financing decisions making can be illustrated with the following example. Looking at the interaction between two agents, financier and entrepreneur, the investment in REE can be schematically separated into three moments. At a first stage ($n = 0$), the technical–economic design of the REE plant is conducted, and its development schedule is determined. At this moment, the entrepreneur must use their own resources to complete the necessary documentation. If the investment is made on a project finance (PF) basis, the expected performance of the project is the main information the financier needs. Therefore, the projected cash flows need to be thoroughly examined [69] and the NPV computed.

In a second step ($n = 1$), the key performance variables are identified and assessed for uncertainty, and the true value of the project can be calculated using the SNPV. If the project is not feasible (*SNPV* negative), both agents abandon the project. In case of continuity, the financier provides the necessary resources for the entrepreneur to start the construction of the plant.

In a third moment ($n = 2$), after construction and before the power plant starts operating, agents can again evaluate the decision to proceed with the project, now based on updated information and less uncertainty.

Figure 4 presents the project stages from the entrepreneur's perspective. Although the entrepreneur performs the first action, i.e., to present the project for financing, it is only after a (possible) proposal from the financier for the interest rate and the resulting evaluation of the SNPV that the first decision moment occurs ($n = 1$): to start the project development, building the plant, or to reject the proposal and abandon the project. In the second step ($n = 2$), after the construction of the plant, if market conditions make the pre-agreed performance of the project unfeasible, the entrepreneur may choose to abandon the project, taking back any residual values, or proceed with the operation of the REE plant.

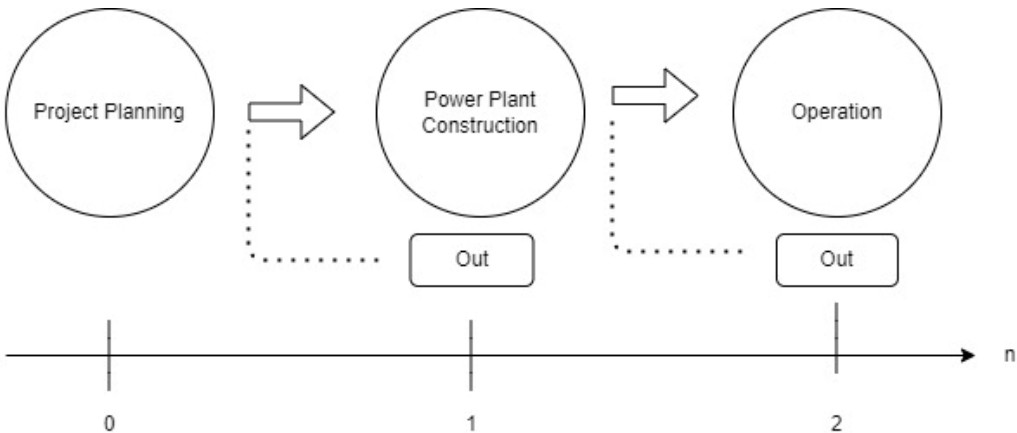

**Figure 4.** Project stages (entrepreneur). Source: Own Elaboration.

For the financier, after analysing the project, the first decision ($n = 1$) is to offer or not the financing. After that, the lender will finance the capital expenditure (capex) in case the entrepreneur accepts the financing. The second decision ($n = 2$) of the lender, after construction, is to continue (or not) with the financing of the operational expenditure (opex), as shown in Figure 5. For both agents, as shown in Figures 4 and 5, the abandonment option may include the recovery of part of the invested amount, through the liquidation of the remaining assets.

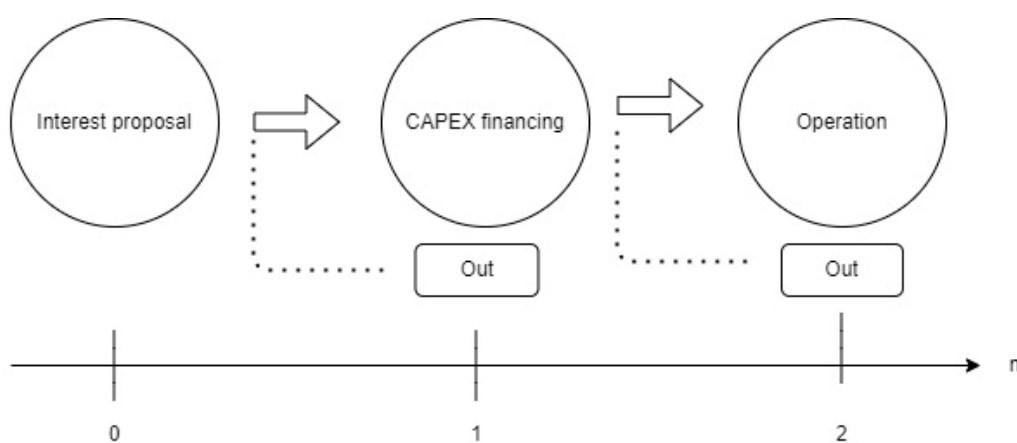

**Figure 5.** Project stages (Bank). Sources: Own Elaboration.

In a generic way, Figure 6 presents an example of the relation between the project's cash flow and the agents' decisions. The depreciation of the financed value will only occur in the period when the plant is already operating, as usual in Project Finance (FP). Nevertheless, the investment should occur at the start of the project, with the amount corresponding to CAPEX. During project execution, in case both parties decide to continue the project, the resources destined to OPEX should still be spent. Access to tranches of funding occurs according to the achievement of pre-agreed performance levels (key performance indicators or KPIs) [70].

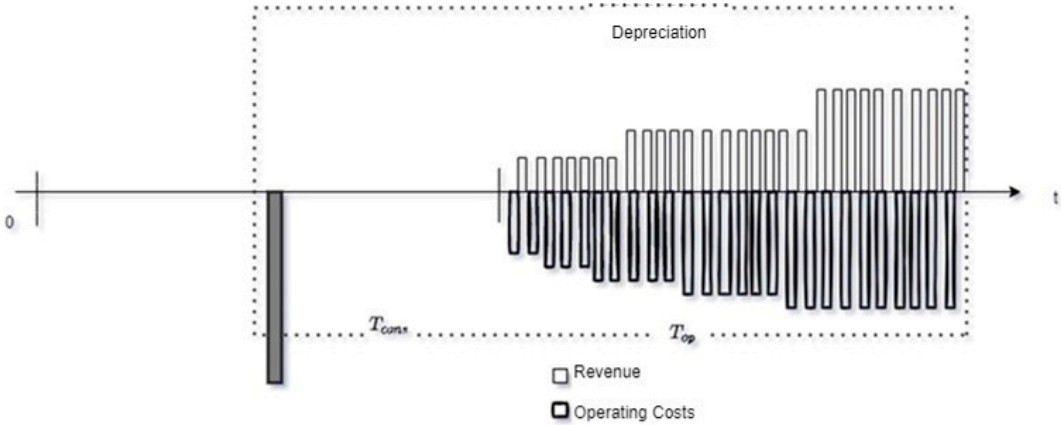

**Figure 6.** Example of a simplified cash flow. Sources: Own Elaboration.

According to Yescombe [71], a project finance has two main phases: construction and operation. As exemplified in Figure 6, after the financing is agreed upon, the entrepreneur has a certain number of periods dedicated to the construction of the plant, and, at the end of this period, both agents must make the decision to abandon and continue the project. Once the decision to continue is taken, the REE plant has an expected number of useful life periods.

## 3. Proposed Model Synthesis

Given the context and the project evaluation methodology presented in the previous sections, this section provides the synthesis of the real options (RO) model in REE projects. The application of RO requires: (1) the initial definition of a moderate scenario used to apply the valuation method upon and estimate the option value of the project; (2) a binomial lattice model applied in the context of renewable electrical energy REE; and (3) the use of discounted cash flow (DCF) or net present value (NPV).

### 3.1. Model Purpose

The purpose of this model is to present solutions to the shortage of public and private funding in terms of green electricity. We focus on the potential for the application of exit options for renewable energy projects in developing countries such as Brazil. The RO approach has been applied in wind, solar, and hydro energy projects in various countries (see Table 1). However, we have no information about the application of RO in the case of Brazil, likely due to the absence of this kind of instrument in the standard credit models proposed by BNDES [9]. Here, we apply the real options analysis framework proposed in Kim [13] and briefly discuss the potential of exit options to boost REE financing. We focus on Brazil, where almost seven thousand publicly funded infrastructure projects (with a total contract value of R$9.32 billion) were suspended between 2012 and 2021 (among those there were renewable energy projects that were paused in the mid of the project execution, probably for insufficient funding, or changes in expected outcomes). This highlights the need to review the way projects are evaluated and approved [72].

Considering the increased uncertainty in a developing country, it seems that a deeper exploration of the RO alternative can enable financing a group of projects that otherwise would not be viable. The great potential is to use these variables as parameters to understand the volatility of the project and define the cost of this volatility through the option value.

Our contribution is to propose a simple model to evaluate projects by considering variables that can reflect on the NPV performance of an infrastructure project and on the country's business environment. We focus on four variables: wages, labour productivity, capital utilisation, and service tariff. As a result, we incorporate volatility arising from the labour market and the consumer energy supply-and-demand market in the valuation model. The idea is that while higher labour productivity can reduce the costs and make the project NPV more effective, higher growth in wages in comparison to labour productivity may offset the positive outcome of efficiency. We use wages and labour productivity to exemplify typical sources of project volatility and define the implicit 'cost' of this volatility through an exit option value. For completeness, we also consider two additional variables, service (energy) tariffs and capital utilisation (service/energy demand), in our model, as we expect that the higher these variables are the higher the supply but also the higher the volatility of the project.

### 3.2. Model Assumptions

As standard practices for project evaluation and analysis process, the projected cash flow is estimated and first evaluated. Equations (1) and (2) represent the cash flow of a simplified REE project, i.e., the cash inflows and outflows throughout its execution, considering discrete time $t = 0, 1, 2 \ldots T$. The only input considered is the operational revenue by time period, i.e., the value obtained when executing the project. As for the cash outflows, we have the investment (CAPEX) in the construction of the plant and the operational cost (OPEX), referring to the expenses in its operation.

In this model, operating revenue $S_t^e$, in each period $t$ is determined by three variables: the tariff (price) per unit of energy sold $p_t^e$ and the plant's installed capacity $K_e$ and its level of utilization $u_t^e$, according to Equation (1). Among the three variables, only the installed capacity is fixed. The tariff $p_t^e$ and utilization $u_t^e$ are defined based on supply and demand in the electricity market.

$$S_t^e = p_t^e \, K_e \, u_t^e \tag{1}$$

The operating cost (OPEX) $C_t^e$ in each period $t$ is determined from the average wage $w_t$, labour productivity $A_e$, and the installed capacity $Ke$ (see Equation (2)). The operating cost $C_t^e$ does not vary with the level of utilization $u_t^e$ of the plant, and $c_t^e$ expresses the unit fixed cost per unit of installed capacity. For simplicity, we consider the unit variable cost to be zero. The average wage $w_t$ follows the dynamics of the labour market. Labour productivity,

on the other hand, is initially estimated ($Ae$ is initially estimated ($e$), but it becomes known (and kept constant) at the beginning of the project's operation.

$$C_t^e = c_t^e K_e \qquad c_t^e = \frac{w_t}{A_e} \tag{2}$$

It is important to point out that the development of an energy plant assessment of future conditions embeds uncertainty at its core. Thus, we assume four cash flow variables as uncertain: the tariff (price) $p_t^e$ the utilization of the plant $u_t^e$ plant utilization, workers' salaries $w_t$ and the plant's productivity $Ae$. The uncertainty amplifies the investment risk for both the financier and the entrepreneur. That is why, as De Salles [73] explains, the financial risk should always be evaluated, and, as Porter [74] proposes, this can be performed adopting future scenario exploration.

Amongst the different approaches to conduct a scenario analysis, Ribeiro [75] highlights the Godet's method. Godet's method is characterized by a morphological analysis of variables and of the most important future-bearing facts. As shown in Table 2, the expected scenarios may be presented in the format of a matrix, thus presenting the intrinsic risks of every variable in our model. From this technique (i.e., the three-point estimation technique), three scenarios are proposed for every variable: the best, the worst, and the moderate [76]. The moderate scenario is usually obtained by the historical moving average of the variables. The extreme and symmetric scenarios are defined from the parameters $\delta_p$, $\delta_w$, $\delta_u$, and $\delta_A$, which represent the expected amplitudes of uncertainty.

**Table 2.** Scenarios of evaluation of the expected values of variables with uncertainty in the power plant project REE.

| Variables | Scenarios | | |
|:---:|:---:|:---:|:---:|
| | **Best** | **Moderate** | **Worst** |
| Tariff | $(1 + \delta_p)\widetilde{p_e}$ | $\widetilde{p_e}$ | $(1 - \delta_p)\widetilde{p_e}$ |
| Salary | $(1 - \delta_w)\widetilde{w_e}$ | $\widetilde{w_e}$ | $(1 + \delta_w)\widetilde{w_e}$ |
| Use | $(1 + \delta_u)\widetilde{u_e}$ | $\widetilde{u_e}$ | $(1 - \delta_u)\widetilde{u_e}$ |
| Productivity | $(1 + \delta_A)\widetilde{A_e}$ | $\widetilde{A_e}$ | $(1 - \delta_A)\widetilde{A_e}$ |

Source: Information collected from (Kim [58]).

Once estimated, the cash flow makes it possible to determine the present value of the investment through NPV analysis. Equation (8) presents the expected NPV of our project model, assuming that the payment for construction is executed only on project's completion:

$$NPV = -\frac{K_e\, c_e}{(1 + r)^{Tcons}} + \sum_{t=Tcons+1}^{Tcons + Top} \frac{\Pi_t}{(1 + r_{deb})^t} \tag{3}$$

In short, the NPV shows the operating result of the enterprise and the difference between operating revenues and costs in the period $t$. In Equation (3), the variable $T_{cons}$ represents the duration of construction, and $T_{op}$ is the plant's operational life. $K_e$ is the installed capacity and $c_e$ is the expected unit cost. The variable $r$ expresses the risk-free interest rate, whereas $r_{deb}$ is the contracted interest rate for the project (WACC).

To estimate the volatility of the project's profitability, we first need to evaluate its profitability under extreme conditions. Equation (4) represents the present value of operating income under the best-case scenario, whereas Equation (5) represents the present value of operating income under the worst-case scenario. The first term in the numerator of both equations consists of the highest (lowest) expected revenue, i.e., the multiplication of the average expected prices and utilisation in the best (worst) case scenario. The second term

in the numerator consists of the projection of the lowest (highest) average expected cost, i.e., the lowest (highest) wage and the highest (lowest) productivity.

$$\Pi_t^M = \sum_{t=1}^{T_{op}} \frac{(1+\delta_p)\, \widetilde{p}_e \cdot (1+\delta_u)\, \widetilde{u}_e \cdot K_e - \frac{(1-\delta_w)\widetilde{w}_e}{(1+\delta_A)\widetilde{A}_t^e} \cdot K_e}{(1+r_{deb})^t} \tag{4}$$

$$\Pi_t^P = \sum_{t=1}^{T_{op}} \frac{(1-\delta_p)\, \widetilde{p}_e \cdot (1-\delta_u)\, \widetilde{u}_e \cdot K_e - \frac{(1+\delta_w)\widetilde{w}_e}{(1-\delta_A)\widetilde{A}_t^e} \cdot K_e}{(1+r_{deb})^t} \tag{5}$$

As previously mentioned, from the extreme expected operating results, and if the project's operating result has a log-normal distribution, we can obtain the expected variance $\sigma^2$ from Equation (6).

$$\sigma_t^2 = \frac{\log\left(\Pi_t^M - \Pi_t^P\right)}{\sqrt[4]{T_{op}}} \tag{6}$$

Considering the objectives of the agents (entrepreneur and financier) and the proposed evaluation methodology, we will go on to propose a (extremely) simplified model of the agents' decision process. At the beginning of the project ($m = t = 0$), the entrepreneur proposes the project to the lender only if the condition of positive expected SNPV is met ($SNPV_0 > 0$) (Equation (11)) and based on the expectations of the interest rate $\widetilde{r}_{deb}$ that would be practiced by the financier. Based on the assumption of a log-normal distribution of risks, the factors $\phi_u$ and $\phi_d$ are calculated according to the equations:

$$\phi_u = e^{\sigma \sqrt{\Delta t}} \tag{7}$$

$$\Phi_d = \frac{1}{\phi_u} \tag{8}$$

where the factor $\phi_u$ describes the positive variation in the asset value per period, and the factor $\phi_d$ the negative variation. From the risk-free interest rate $r$ (or the interest rate $\widetilde{r}_{deb}$), the risk-neutral probability $q$ can be approximated by Equation (9).

$$q = \frac{e^{r\,\Delta t} - \phi_d}{\phi_u - \phi_d} \tag{9}$$

Thus, the option value is calculated using Equation (10). The variable $V_n$ is the option value that is obtained recursively (Figure 3) by discounting the risk-free rate $r$ weighted by the risk-neutral probability $q$ from the value of subsequent nodes $V_{n+1}^{u,d}$:

$$V_n = e^{-r\,\Delta t}\left[q\, V_{n+1}^u + (1-q)V_{n+1}^d\right] \tag{10}$$

The strategic net present value *SNPV* is determined as follows:

$$SNPV = NPV + V_0 \tag{11}$$

where Net present Value (NPV) is the difference between revenues, investments, and costs discounted by the WACC. $V_0$ represents the value of the option to abandon the project.

In the following period ($m = t = 1$), the entrepreneur reassesses the Strategic net present value (SNPV) of the project's realization with the interest rate $r_{deb}$ offered by the bank, and they proceed with the construction of the plant if the value is positive ($SNPV_1 > 0$) under the current conditions of the energy and labour markets (Tariff, demand (usage) and wage values may have changed since period 0 altering the scenario). In this case, the bank provides the first part (tranche) of the financing to cover the capital costs.

In the next step ($m = 2$, $t = T_{cons} + 1$), after construction of the plant, the entrepreneur proceeds with the project if and only if the operating NPV is positive under current market conditions. It is worth noting that the initially agreed-upon rate ($r_{deb}$) may no longer be adequate for the bank under market conditions as interest rates vary.

## 4. Model Application to the Brazil Case of Itumbiara

The electricity sector in Brazil is divided into generation, transmission, distribution, and commercialization. In a simplistic way, generators produce energy, transmitters transport it from the point of generation to substations in large consumer plants, and distributors take it from there to citizens' homes and companies. The Brazilian energy sector is made up of public power companies and institutions (e.g., the National Agency for Electrical Energy (ANEEL), Eletrobrás, and the Energy Research Company (EPE)) and private initiatives that operate on different fronts, from generation to distribution, including the regulation of the sector. ANEEL is responsible for (i) implementing the federal government's policies and guidelines for the exploitation of electric power and hydraulic potentials and (ii) regulating the granted, permitted, and authorized services by issuing the necessary regulatory acts [77,78]. The purpose of the EPE is to provide research services to the Ministry of Mines and Energy (MME) to support the planning of the energy sector, covering electricity, oil, and natural gas and their derivatives and biofuels [79]. ANEEL organizes and approves the energy auctions held to contract the purchase of electricity by delegation and in accordance with the guidelines of the Ministry of Mines and Energy [77].

We consider the project for a plant such as the Itumbiara Hydroelectric Plant, the largest plant in the Furnas System (Brazilian hydroelectric power plant systems with facilities in the states of São Paulo, Minas Gerais, Rio de Janeiro, Espírito Santo, Paraná, Goiás, Mato Grosso, Mato Grosso do Sul, Pará, Tocantins, Rondônia, Rio Grande do Sul, Santa Catarina, Ceará, Bahia, and the Federal District). In terms of generation potential, the Itumbiara Power Plant is considerably smaller than the Itaipu Power Plant, one of the largest power plants in the world, being able to generate around 85% less than the Itaipu plant [80]. The total investment of the project (construction) is USD 187,589,100, and it is expected to last for 47 years. The first seven years were dedicated to construction and the remaining 40 years to operation (concession period). We consider the same periods in this simulation. Table 3 summarizes the expected values for the project (moderate scenario). There is no precise information on how long of a time was needed for the planning and deduction of the project's finances. Therefore, this assessment assumes that the project finance has already been prepared and is ready for being evaluated by the investing agent. The information regarding the necessary investment amount, as well as schedules and the other variables presented in Table 3, were obtained through official institutional channels, such as ANEEL [33,81], BB [8], BNDES [9,42], and EPE [79].

**Table 3.** Example design data for a hydropower plant (moderate scenario), values in US dollars.

| Description | Values |
|---|---|
| Installed generation capacity ($K_e$) | $2.3994 \times 10^6$ MWh |
| Unit investment cost ($ci_e$) | 187,589,100.00 million USD |
| Construction period ($T_{cons}$) | 7 years |
| Operating period ($T_{op}$) | 40 years |
| Risk − free interest rate ($r$) | 4.5% |
| Interest rate contracted by the entrepreneur ($r_{deb}$) | 8% |
| Expected average unit operating cost $\left( \tilde{c}_e \right)$ | 0.13 USD/kWh |
| Expected average electricity tariff $\left( \tilde{p}_e \right)$ | 0.26 USD/kWh |
| Expected average salary $\left( \tilde{w}_e \right)$ | 0.89 USD/ht |

**Table 3.** *Cont.*

| Description | Values |
|---|---|
| Expected average productivity $\left( \widetilde{A}_e = \frac{\widetilde{w}_e}{\widetilde{c}_e} \right)$ | 0.89 USD/ht/0.13 USD/kWh = 6.84 kWh/ht |
| Average expected usage $\left( \widetilde{u}_e \right)$ | 53% |

Legend: kWh = Kilowatt-hour; ht = working hours; Source: Prepared by the authors based on ANEEL [33,81], BB [8], BNDES [9,42], and EPE [79].

The expense incurred in the planning phase does not make up the amount required for financing, and it is disregarded for simplicity. To analyse the feasibility of the project, it is paramount that uncertainty ranges are identified, as per Table 4, and that projected cash flows are computed for the three scenarios (worst, best, and moderate).

**Table 4.** Evaluation scenarios for the average expected values of the varieties with uncertainty in the example plant design REE.

| Variables | Scenarios | | |
|---|---|---|---|
| | **Best** | **Moderate** | **Worst** |
| Tariff $\left( \widetilde{p}_e \right)$ | 0.88 USD/kWh (128%) | 0.69 USD/kWh (100%) | 0.49 USD/kWh (72%) |
| Salary $\left( \widetilde{w}_e \right)$ | 0.74 USD/ht (84%) | 0.89 USD/ht (100%) | 1.03 USD/ht (116%) |
| Usage $\left( \widetilde{u}_e \right)$ | 61% (115%) | 53% (100%) | 45% (85%) |
| Productivity $\left( \widetilde{A}_e \right)$ | 7.93 kWh/ht (116%) | 6.84 kWh/ht (100%) | 5.74 kWh/ht (84%) |

Source: Elaborated by the author.

From the three-point estimation technique, intervals were established for every variable. According to ANEEL [33,81], the energy tariff in Brazil between the years 2010 and 2021 had an average value of 0.69 USD/KWh, with an average variation of 28%. According to IBGE [82], the salary (Brazilian minimum wage) had an average value of 0.89 USD/ht and maximum variation of 16% between the years 2012 and 2021. The use of physical capital in the moderate scenario was defined from the relation between the average of what was generated annually in electric power at the Itumbiara plant in recent years and the annual projection of the plant's total electric power generation capacity. The productivity in the moderate scenario was established from Table 3.

As previously explained, the operating profit in every period is equal to the operating revenue less the operating cost in that period. Considering that the Fiscal Year Income Statement (DRE) is not ready during project execution, the best possible estimate to calculate the periodic revenue is through the calculation of the average expected revenue by subtracting Equation (2) from Equation (1).

$$\widetilde{\Pi}_t = \widetilde{S}_e - \widetilde{C}_e = \left( \widetilde{p}_e \cdot \widetilde{K}_e \cdot \widetilde{u}_e \right) - \left( \frac{\widetilde{w}_e}{\widetilde{A}_e} \right) \widetilde{K}_e$$

Evaluating the project development only by the NPV (Equation (8)) and making use of the values proposed in the moderate scenario in Table 3, we obtain the following:

$$NPV = \frac{-(2.3994 \times 10^6 \cdot 0.13)}{(1 + 0.045)^7} + \sum_8^{47} \frac{(0.69 \cdot 2.3994 \times 10^6 \cdot 0.53) - \left( \left( \frac{0.89}{6.845} \right) \cdot 2.3994 \times 10^6 \right)}{(1 + 0.08)^t}$$

Even with the simplifications used, the result (452,382.11 USD) indicated that the project would be considered feasible. However, if the decision to pursue the project (considering the value of the option to abandon the project) is made according to the NPV analysis, then the situation becomes more attractive. From Equation (6), we have:

$$\sigma_1^2 = \frac{\log\left(1.5027 \times 10^6 - (-450924.81)\right)}{\sqrt[4]{40}} = 6.29$$

Having obtained the project variance from Equations (1) and (2), we obtain $\phi_u = 1.87$ e $\phi_d = 0.53$ e $q = 0.34$. Considering an initial investment of USD 187,589,100.00, the project and option valuation lattices can be computed, as in Figure 7.

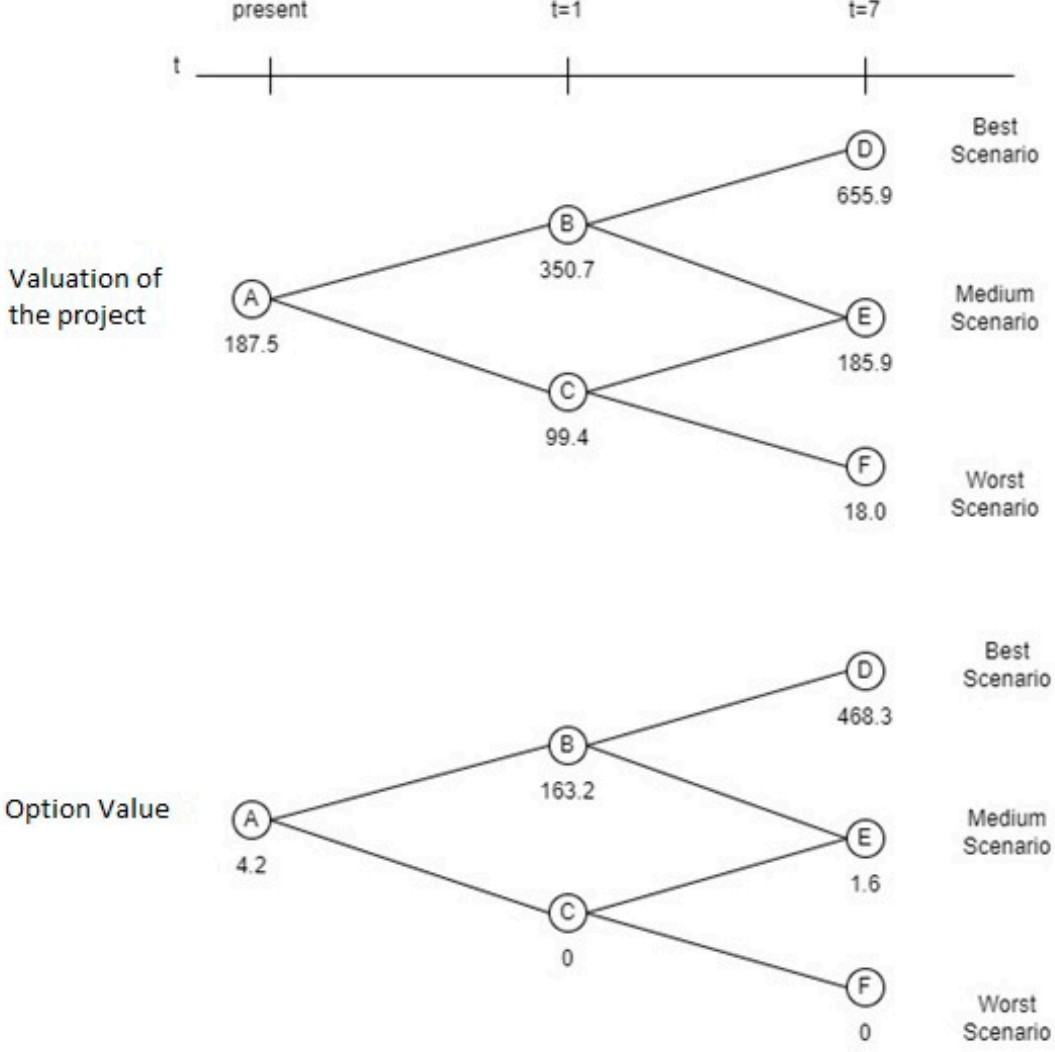

**Figure 7.** Project and option valuation lattices (in USD million, rounded values). Source: Elaborated by the author.

Figure 7 presents the option value where a decision for proceeding or abandon the project was not made at the start of the project (t = 0), as $V_0 = 4.26 \times 10^6$ USD obtained from Equation (10), i.e., the probability of neutral risk was determined using Equation (9). $q = 0.34$. Thus, from Equation (11), we have $SNPV = 1.12 \times 10^7$, increasing the present value of the project by 61%.

*Sensitivity Analysis*

To perform the sensitivity analysis, the variables' initial investment, risk-free rate of return, interest rate contracted by the entrepreneur, and plant operation time were kept as constants. For the variables $p_e, w_e, u_{e,}$ and, $A_e$, established as parameters, and their variations for the best and worst scenarios, we used the parameters as a uniform distribution. The time interval between decisions, i.e., $\Delta t$, was also established as a random result of a uniform distribution, where $2 < \Delta t < 10$.

The model is simulated one hundred thousand times using Monte Carlo analysis. By the Law of Large Numbers, the expected value of a random variable can be approximated from an empirical average of independent samples of variables. In this sense, the more an experiment is repeated the more precise the estimation of the probability of an event to occur [83].

Figure 8 shows the calculated option values (in logarithm) in relation with the estimated volatility. The size of the circles gets larger as the $\Delta t$ increases, where $\Delta t$ r is the time interval between the agents' decisions to continue or discontinue the project. Red values are calculated when the probability q is less than 0.5, and blue values are calculated when the probability q is greater than or equal to 0.5. As the figure shows, most projects have a probability of becoming more profitable below 0.5 within the ranges of our simulations. This suggests that a plant such as the Itumbiara Hydro Plant has a lower probability of achieving economic viability using ex-ante analysis, even if the main driver for this assessment is exclusively connected to the historical volatility embedded in the main variables. In many cases, perfectly feasible projects are discarded because of valuation methodology constraints on dealing with the uncertainty associated with long-term projects using ex post analysis.

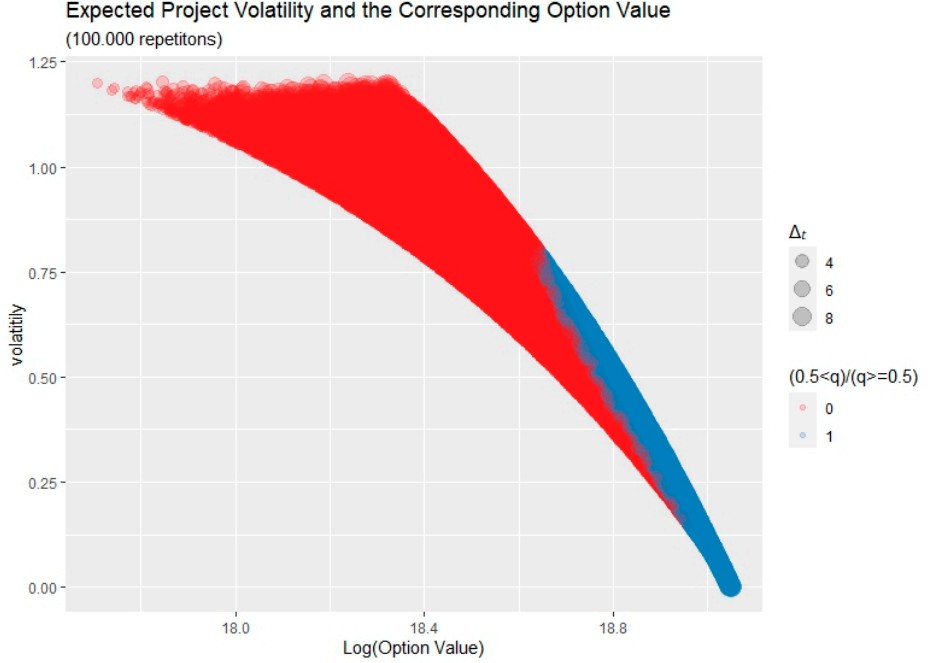

**Figure 8.** Expected Project Volatility and the Corresponding Option Value (USD million/100 M repetitions). Source: Elaborated by the author.

Regarding the variable wage, Figure 9 compares the worst-case scenario (see the box plots at the top) with the best-case scenario (see the box plots at the bottom). As it is possible to see, the central lines indicate the median data between the two and the fact that there were no substantial differences, i.e., we see that there are no substantial differences when we consider the probability $q$ as the reference factor by looking at the median lines,

although the scales are different. Similar behaviour was observed for the productivity and capital utilization variables.

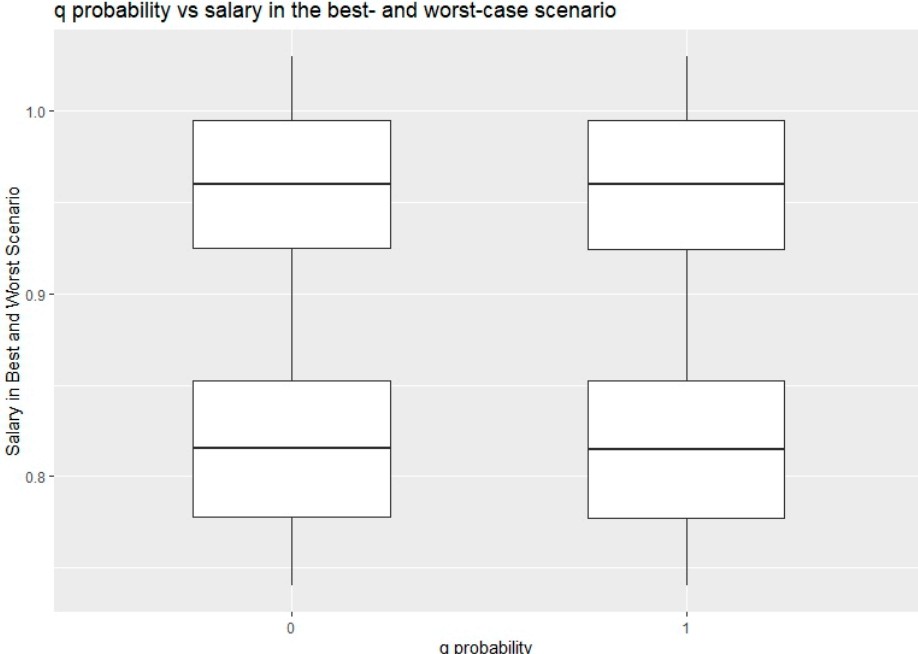

**Figure 9.** Probability q against salary in the best- and worst-case scenario. Source: Elaborated by the author.

However, unlike the variables mentioned above, the energy tariff levels (price) in the worst-case scenario are higher when probability q is higher (see Figure 10). Figure 11, however, shows the opposite, i.e., larger tariffs when probability q is smaller in the best scenario. In doing so, the energy tariff has a pro-cyclical behaviour in the worst scenario and an anticyclical one in the best one.

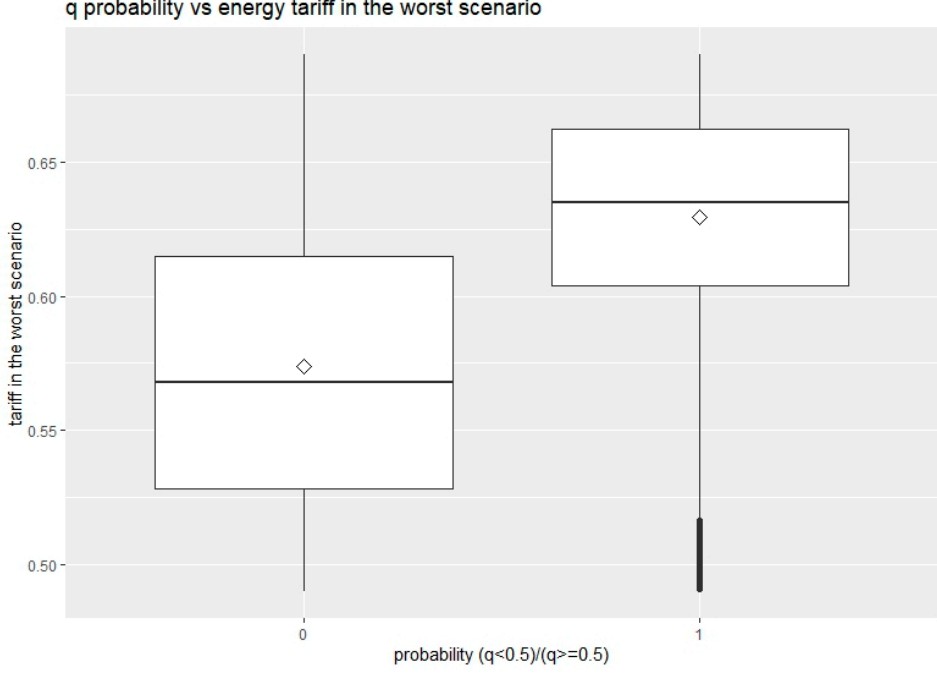

**Figure 10.** Probability q versus energy tariff in the worst scenario. Source: Elaborated by the author.

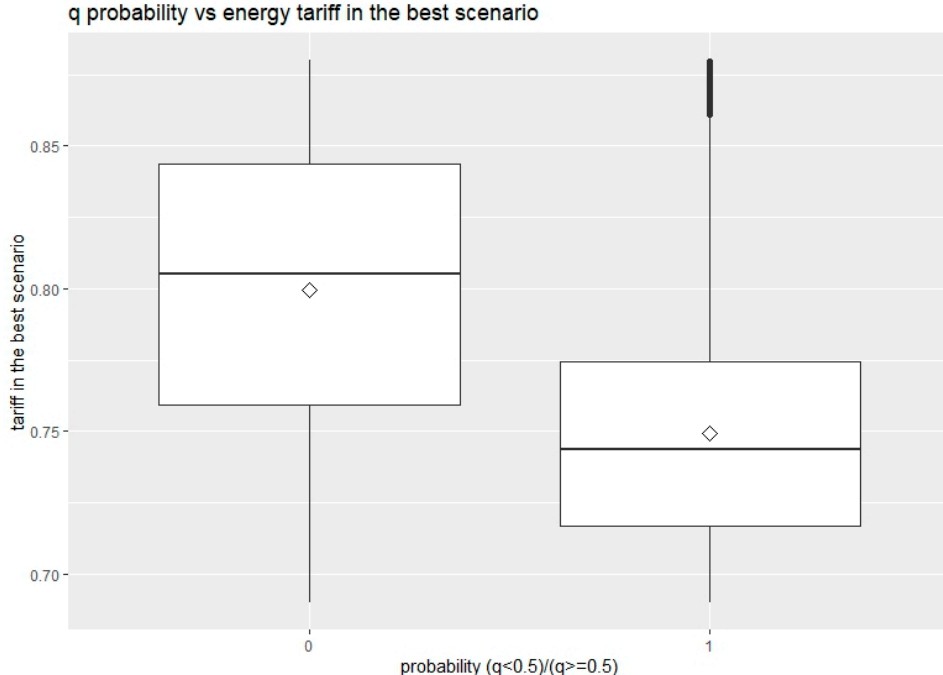

**Figure 11.** Probability q versus energy tariff in the best scenario. Source: Elaborated by the author.

Finally, Figure 12 shows the distribution of calculated option values with respect to probability q. Therefore, there is a certain point of intersection where a probability q becomes greater or lower than 0.5 and returns a similar option value. In this range, agents would obtain a similar option value even with a different market volatility.

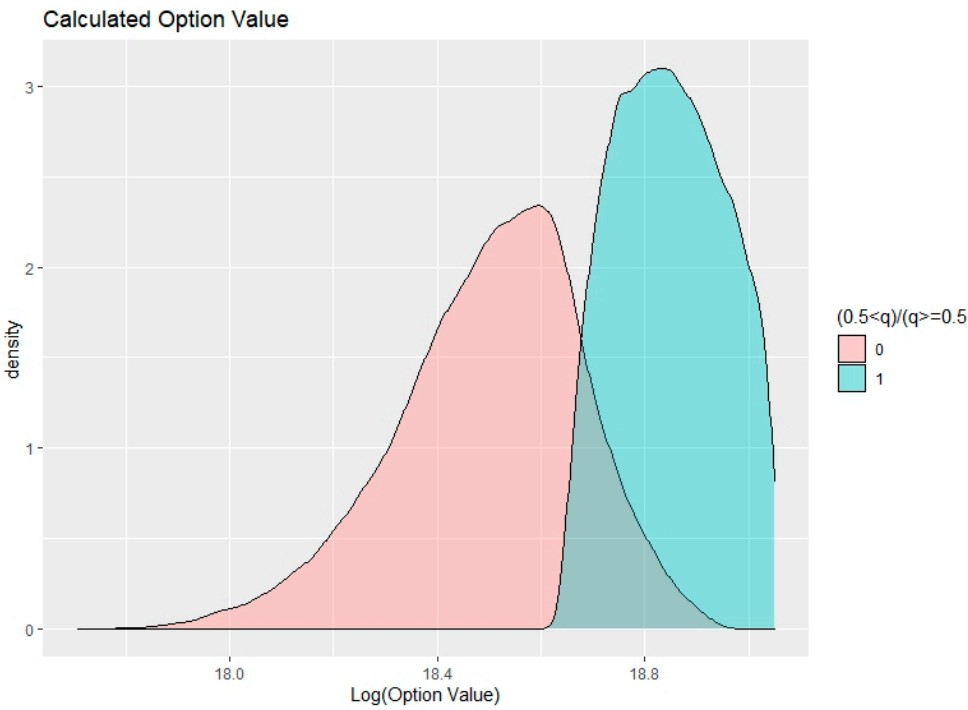

**Figure 12.** Calculated option value. Source: Elaborated by the author.

## 5. Discussion of Results

The experimental results proposed in this paper appear to be consistent with the low levels of private investment in the infrastructure sector in Brazil and many developing countries. The documented higher volatility of domestic markets in these economies, plus

the limited ability of the usual valuation methods—developed for advanced countries—to deal with such level of uncertainty, seem to indicate that new approaches as the one proposed in this paper are needed in order to support a satisfactory level of engagement from private finance.

The model validation process showed that a project investment such as renewable electric power generation becomes more financially attractive when the real options analysis method is applied, as argued by Kjaerland [14], i.e., despite the assumption of a positive NPV, the projects show greater profitability when considering the option value. However, these projects share different probabilities of success. Furthermore, this study complements the work performed by Zavodov [16] by showing how the application of real options analysis, especially in hydroelectric projects, is also applicable for developing economies such as Brazil.

Assuming that Brazil is a country that seeks to encourage the use of renewable energy sources, with targets for the use of these sources and fiscal incentives, the evaluation of energy projects such as Itumbiara's from the RO perspective could be another factor encouraging greater private sector participation [17,21,84]. In other words, as we become more effective in this type of financing, there is a greater possibility of greater engagement of private companies in renewable energy projects. However, it is worth noting that this is a completely speculative observation and requires further discussion. Comparing the results obtained with the results obtained by Kim [13] (that differ from what is proposed in this paper), who used the Certified emission reduction (CER) variable without considering wages and labour productivity in their study, the volatility of the project profitability was higher. For this reason, the probability $q$ was much lower in the case study applied in this work than in the one obtained by Kim [13]. Finally, considering that the power generation market is inserted in a complex system where other agents also seek access to finance for their projects, it becomes interesting to evaluate the results of this model in a context of complexity economics and policy (see an example of a multisectoral model [85]).

## 6. Conclusions

This paper evaluated an exit-option model applied to the finance of large projects, such as renewable electrical energy (REE) generation. The proposed model expands the existing real options analysis (ROA) literature by considering uncertainty associated with cost-side variables such as wages and labour productivity. It is a tool to evaluate the overall valuation volatility of large-scale projects, considering the value of strategical opportunities after they are started. This is particularly important in the case of REE projects, which are exposed to substantial uncertainty in the attempt of forecasting many of the variables required for financial valuation. In this scenario, projects that are perfectly viable ex post may be rejected ex ante due to disregarding the real value of strategical options not included in traditional valuation models, such as the early abandonment of the project. This is particularly important if one wants to engage the private sector in financing REE projects in developing countries, where uncertainty about the future is always higher than in industrialized regions, under a scenario of urgent need for energy transition. The ROA methodology has been used in recent years as a complement to traditional project valuation approaches, and the current study is an expansion of the ROA with the aim of adding some key variables (wages and labour productivity) to the analysis and with focus on developing countries. We expect that our contribution may indicate some possible alternatives to accelerate the green energy transition.

To make our contribution more tangible, particularly for emerging countries, we applied the proposed ROA-augmented methodology for the analysis of an existing hydro power project in Brazil. The country, despite a mostly sustainable energy-generation matrix, still needs REE sources to supply the ever-growing demand for electricity. The model application to this case showed that the proposed methodology would have increased the ex ante financial attractivity of the chosen project substantially. The consideration of new variables such as wages and labour productivity resulted in a significantly higher

volatility of the project valuation, an important finding, especially for developing countries. The possibility of agents to "neutralize" the uncertainty by considering the value of future strategic options, obtaining a viable project valuation even under a higher market volatility, can be useful as a highlighted early-exit option.

Despite the adequacy shown by the model in the simple application proposed here, it still relies on simple assumptions with limited applicability for supporting policy decisions. From the perspective of supporting policymakers in using systems thinking and complexity, in view of the UN conferences of parties' (COP) events on climate change, we propose to incorporate the augmented-ROA conditions in an agent-based simulation model able to model both the overall macro dynamic and the specific processes occurring in the energy sector: in particular, the competition between renewable and carbon-based energy generation. Beyond COP28, we highlight that COP30 will be hosted exactly in Brazil, and further development of this line of research may prove fruitful to be proposed at those events.

**Author Contributions:** Conceptualization, A.C.M. and M.d.C.P.; methodology, A.C.M.; software, A.C.M.; validation, A.C.M.; formal analysis, A.C.M.; investigation, A.C.M.; resources, A.C.M.; data curation, A.C.M.; writing—original draft preparation, A.C.M., M.d.C.P. and R.P.; writing—review and editing, A.C.M., M.d.C.P. and R.P.; visualization, A.C.M.; supervision, M.d.C.P. and R.P.; project administration, M.d.C.P. and R.P.; funding acquisition, R.P. and M.d.C.P. All authors have read and agreed to the published version of the manuscript.

**Funding:** This research was funded by the Coordenação de Aperfeiçoamento de Pessoal de Nível Superior (CAPES)-Brazil, Funding Code 001, the Fundação de Amparo à Pesquisa do Estado de São Paulo (FAPESP), process no. 2015/24341-7, the Department for Energy Security and Net Zero (DESNZ) and the Children Investment Fund Foundation (CIFF)–United Kingdom, project consortium Economics of Energy Innovation Systems Transition (EEIST).

**Institutional Review Board Statement:** EEIST is jointly funded through UK Aid by the UK Government's Department for Energy Security & Net Zero, and the Children's Investment Fund Foundation (CIFF). Contributing authors are drawn from a wide range of institutions. For full institutional affiliations see www.eeist.co.uk. The contents of this paper represent the views of the authors and should not be taken to represent the views of the UK government, CIFF or the organisations to which the authors are affiliated, or of any of the sponsoring organisations.

**Informed Consent Statement:** Not applicable.

**Data Availability Statement:** The Data and the model used for this study are available at the link https://github.com/annacarolmartz/ROA_Monte_Carlo_Simulation_BRAZIL (accessed on 3 June 2023).

**Acknowledgments:** The authors would like to thank the reviewers of this paper who supported improving it and make a stronger contribution.

**Conflicts of Interest:** The authors declare no conflict of interest.

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
