# Peer review of "Renewable Electricity Transition: A Case for Evaluating Infrastructure Investments through Real Options Analysis in Brazil"

_sustainability, doi:10.3390/su151310495_

Round 1

Reviewer 1 Report

This paper proposes a real-option analysis method to evaluate the profitability of renewable power generation projects considering different sources of uncertainties such as wages and productivity. The benefits of the proposed method have been assessed against other traditional deterministic approaches. This reviewer has the following comments and suggestions to improve the presentation and clarity of the paper:

-          Please clarify explicitly and precisely the sources of uncertainties under consideration. More important, how they are considered in the modelling.

-          Please better define the adopted financial metrics to assess the performance of the proposed method. What is the planning horizon? How the investments have been defined? Please clarify how the electricity tariff is found/calculated?

-          Please better describe the energy regulations in Brazil and how do the current and the future policies are considered in the proposed method.

-          Please re-write the conclusions to better highlight the key insights of the manuscript.

-          Please provide all the abbreviations in Table 1.

-          Please check the text flow in Section 2.4. Is it better to move the equations to different Section?

-          Please reproduce Fig.6. It is not clear.

-          What is the kWh/ht? Please define it.

Author Response

Por favor, verifique o anexo.

Reviewer 3 Report

The paper focuses on analyzing variables such as wages and labour productivity to evaluate project volatility (in terms of profitability) by using real options analysis (ROA). I think the paper can answer the big question of why renewable energy projects were suspended due to insufficient funding or changing the expected outcomes. In Table 1, real option analysis methodologies applied to renewable energy projects are listed clearly. However, in Section 3, the authors only focus on analyzing the NPV method without discussing the ROA, which is considered the main contribution of the manuscript. Can the authors explain these abnormal points? Moreover, I cannot see the results made by using any ROA methods; almost all the results are simulated from the NPV in Section 4. Section 5 is poorly written because it lacks the comparative results in Section 4 among the valuation methods applied in the Brazil case.

In addition, the keyword part included the "game theory" term. But I cannot see the game theory issue that is discussed in the manuscript. I think the author should rewrite with a big change for this topic.

The English should be improved more to present some points clearly, especially the names of the expected methods.

Round 2

Reviewer 1 Report

Most of the comments raised in the first round of review have been adequately addressed

Reviewer 3 Report

The authors reflected on the comments. Thank you!

The English for describing the comments is clearly inserted. Thank you for your effort!